# ImmersePro: End-to-End Stereo Video Synthesis Via Implicit Disparity Learning

Jian Shi [1]   Zhenyu Li [1]   Peter Wonka [1]

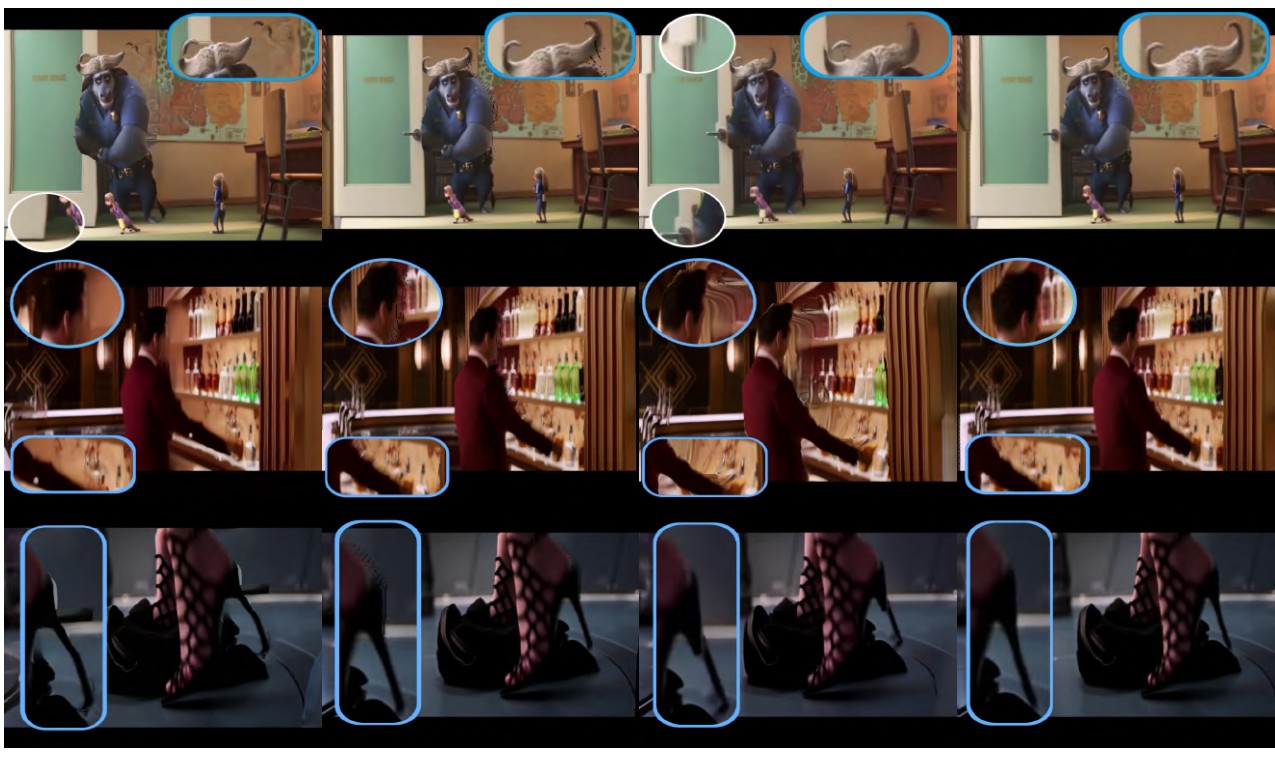

Figure 1. ImmersePro converts a single-view video to a stereo video by predicting plausible right-view images for each input frame. Compared to previous work processing images frame by frame (3D Photo or Stereo from Mono), our method has the best visual quality.

|  |  |  |  |
| --- | --- | --- | --- |
| 3D Photo | Stereo From Mono | Stereo Diffusion | Ours |

## Abstract

We introduce *ImmersePro*, an innovative framework specifically designed to transform single-view videos into stereo videos. This framework utilizes a novel dual-branch architecture comprising a disparity branch and a context branch on video data by leveraging spatial-temporal attention mechanisms. *ImmersePro* employs implicit disparity guidance, enabling the generation of stereo pairs from video sequences without the need for explicit disparity maps, thus reducing potential errors associated with disparity estimation models. In addition to the technical advancements, we introduce the YouTube-SBS dataset, a comprehensive collection of 423 stereo videos sourced from YouTube. This dataset is unprecedented in its scale, featuring over 7 million stereo pairs, and is designed to facilitate training and benchmarking of stereo video generation models. Our experiments demonstrate the effectiveness of *ImmersePro* in producing high-quality stereo videos, offering significant improvements over existing methods. Compared to the best competi-

[1]KAUST, Thuwal, Saudi Arabia. Correspondence to: Jian Shi <jian.shi@kaust.edu.sa>, Peter Wonka <peter.wonka@kaust.edu.sa>.

*Proceedings of the 43rd International Conference on Machine Learning*, Seoul, South Korea. PMLR 306, 2026. Copyright 2026 by the author(s).

tor stereo-from-mono we quantitatively improve the results by 11.76% (L1), 6.39% (SSIM), and 5.10% (PSNR).

# 1. Introduction

A stereo movie, also known as a 3D movie, provides three-dimensional visual effects by employing stereoscopic techniques. By capturing or creating separate views for the left and right eyes, a 3D immersive experience can be achieved by using dedicated hardware such as head-mounted displays or autostereoscopic displays. The disparity between the two views perceived by the viewer's brain creates the illusion of depth, making the objects in the movie appear at varying distances, thereby enhancing the immersive experience of the film. Shooting stereo movies in the film industry often involves high costs due to the need for specialized equipment and meticulous post-production processes. Alternatively, the stereoscopic effect can be created through a post-production process for videos that are shot with monocular cameras. This post-production process uses *stereo conversion*, which adds the binocular disparity depth cue to digital images. It requires significant manual work by artists since inaccurate depth mapping and misrepresentations of occluded areas can cause visual discomfort (Devernay & Beardsley, 2010). In this paper, we propose an automated system that can reduce the time and expense associated with the conversion process, making it more accessible and economically feasible for more films.

Traditional *stereo conversion* involves creating disparity maps from single images or sequences and then using them to generate the corresponding stereo pair for the other eye, creating the illusion of depth for stereoscopic viewing. Recently, many deep learning-based methods (Xie et al., 2016; Wang et al., 2019a; Shih et al., 2020; Watson et al., 2020; Ranftl et al., 2022) are primarily proposed for image-based stereo conversions, aiming to improve disparities and enhance inpainting effectiveness on occluded areas. Unlike image data, video data provides additional temporal information, which can yield more detailed disparities and occlusion insights by leveraging information across frames. To handle video inputs, (Chen et al., 2019) synthesizes right-view video sequences by estimating a displacement map to move each pixel to a new location, with a 3D DenseNet. Temporal3D (Zhang & Wang, 2022) compromises to use three adjacent left-view frames to predict the single right-view of the middle frame. Based on our analysis, current stereo conversion frameworks for video sequences are not robust and have several drawbacks. We believe the area is under-explored and there is a large room for improvement. At the same time, we believe the topic will gain in importance due to recent efforts to manufacture stereo displays, e.g., from Apple and Magic Leap.

We introduce *ImmersePro*, a novel approach designed specif-ically for video stereo conversion that utilizes the contextual information available across video frames to enhance stereo disparity consistency across the temporal dimension. For doing so, we collectively build a large-scale stereo movie dataset, *Youtube-SBS*, with over 7 million stereo pairs from a collection of stereo movies, game films, and music videos. Due to the absence of ground truth disparities, we propose to use implicit disparities to guide the generation of *layered disparities*, which outperforms the explicit disparity guidance (*e.g.* a depth estimation model) that was commonly used in previous work. We propose to use a *layered disparity* representation that refers to a stack of disparity maps corresponding to one image. Each pixel that appears in the image can be reused multiple times, avoiding creating black holes after the warping operation. This approach ensures that the generated stereo parts strictly adhere to the semantics of the input video, minimizing the need for improvisation and thus preserving the original narrative and visual intent. As a result, *ImmersePro* not only maintains the semantic integrity of the original video but also intelligently infers the geometry of occluded areas, enabling consistent right-view generation. As shown in Figure 1, previous methods may generate artifacts such as texture misalignment or object deformation, whereas our *ImmersePro* can keep the semantic integrity from the left-view image. Our main contributions are as follows:

- We introduce the *YouTube-SBS* dataset, an extensive collection of stereo videos sourced from YouTube, featuring over 7 million stereo pairs. This dataset fills the gap to serve as a benchmark for training and evaluating stereo video generation models.

- We introduce *ImmersePro*, specifically tailored for converting single-view videos into stereo videos using *layered disparity* warping via implicit disparity guidance. Compared to the best competitor stereo-from-mono we quantitatively improve the results by 11.76% (L1), 6.39% (SSIM), and 5.10% (PSNR).

# 2. Background

## 2.1. Stereo Conversion Methods

**Image-Based Stereo Conversion.** Deep3D (Xie et al., 2016) relaxes the disparity map into a multi-layer probabilistic map and then multiplies it with several horizontally shifted copies of the input image, which relaxes the non-differentiable warping operation. (Watson et al., 2020) used a warping-and-inpainting framework, which creates stereo training pairs from single RGB images to improve the modern monocular depth estimators. However, a non-differentiable strategy is used and the inpainting randomly selects the texture from the training set. Apart from using pretrained depth estimation models, (Wang et al., 2019a;

Ranftl et al., 2022) use FlowNet2.0 (Ilg et al., 2017) to estimate optical flows as ground truth disparities. StereoDiffusion (Wang et al., 2024) proposes a training-free approach to generate stereo pairs by directly warping the latent space of diffusion models. It requires inversion methods to produce the latents to generate the stereo pair of a given image. The fine details of the resulting photo may vary due to the direct modification of the latent space. (Shih et al., 2020) proposed a layered depth inpainting method that generates a 3D representation by intelligently estimating and filling depth information, particularly in areas where it is missing or uncertain. Our work does not rely on explicit disparity computation, with the additional consideration of the context within video frames.

**Video-Based Stereo Conversion.** (Chen et al., 2019) adopts a reconstruction-based approach by using a 3D DenseNet to estimate the disparity map of an input sequence. *Temporal3D* (Zhang & Wang, 2022) estimates the middle frame using three adjacent frames, with the output being a weighted sum of three disparity-warped images. Additionally, methods such as *NVDS* (Wang et al., 2023) may be adopted for consistent depth estimations across video frames. However, those methods assume the pixels within the left image are adequate for the right image. (Mehl et al., 2024) adopted the warping-inpainting approach with a pretrained depth estimation method (*i.e.* MiDaS (Birkl et al., 2023)) for warping and inpainting with multiple adjacent frames. Still, this method relies on a single frame depth estimation model that can likely break the temporal consistency between frames. In this work, we propose an end-to-end video stereo conversion method based on implicit disparity guidance across the temporal dimension.

## 2.2. Stereo Datasets

There are limited resources on video-based stereo datasets. Sintel (Butler et al., 2012) contains 1064 synthetic stereo images with accurate disparities. KITTI (Menze & Geiger, 2015) offers 8.4K frames captured from the real world for autonomous driving. (Wang et al., 2019a) introduces a *WSVD* dataset and proposes to use optical flow as disparities as ground truth for supervision. Similarly, (Ranftl et al., 2022) collected a private 3D movie dataset and extracted ground truth disparities by estimating optical flows to improve depth estimation. Since different levels of stereoscopic effects may exist for different purposes of a dataset, a movie-specific benchmark dataset is preferable. (Ranftl et al., 2022) is the only relevant dataset, but it is built on top of real movies with intellectual property rights issues. Therefore, we propose a benchmark stereo dataset that contains publicly available content.

*Table 1.* Relevant datasets.

| Dataset | content | GT depth | available | No. frames |
|---|---|---|---|---|
| KITTI (Menze & Geiger, 2015) | autonomous driving | metric | Y | 8.4K |
| WSVD (Wang et al., 2019a) | mixed | NA | Y | 1.5M |
| 3D Movies (Ranftl et al., 2022) | movies | NA | N | 75K |
| Sintel (Butler et al., 2012) | synthetic | metric | Y | 1064 |
| Youtube-SBS | movies | NA | Y | 7M |

## 3. Youtube-SBS

We aim to set up a large-scale publicly accessible benchmark dataset. The direct collection of 3D movies often encounters legal challenges to publishing as an open-source dataset. Therefore, we present *Youtube-SBS*, an open-source dataset collected from YouTube. This dataset contains over 400 3D side-by-side (SBS) videos. With a particular interest in stereo movies, our dataset primarily consists of movie trailers, game films, and music videos. We explicitly excluded 360-degree virtual reality videos and gameplay videos (that contain user interfaces). To ensure accessibility for future research, we select videos that (1) have existed for at least one year, and (2) from accounts that have at least 500 followers. This curated selection includes 423 videos at a standard resolution of 1920x1080. During the frame extraction, as some videos include a non-stereo intro section, we skip the first 600 frames to capture valid stereo pairs.

To measure the general stereo effects of our dataset, we propose to compute a metric that evaluates the left-right consistency of the disparity. For a stereo pair with subtle stereo effects, the disparity maps for the left and right images should be almost symmetrical with one another. That is, a point in the left image should have a corresponding point in the right image at the same row but shifted horizontally according to the disparity. For large stereo effects, there is an increasing number of occluded and disoccluded areas. In these regions, the right image can no longer be reconstructed from the left image with simple warping (and vice versa). To compute our metric, we use the optical flow method RAFT (Teed & Deng, 2020). We also evaluated STTR (Li et al., 2021) and RAFT-Stereo (Lipson et al., 2021), but these two methods produced worse results. Note that high consistency means that the left-to-right optical flow $F_{l \to r}$ and right-to-left optical flow $F_{r \to l}$ are the negative of each other. We calculate the consistency $\varepsilon$ as follows:

$$\mathcal{E}_p = ||F_{l \to r}(p)) + F_{r \to l}(p + F_{l \to r}(p))||, \qquad (1)$$

where $p$ is the pixel position of a frame. We provide a breakdown to demonstrate the consistency metric in Table 2 to present the general stereo effects of the dataset. We compute occluded areas with $\sum_p 1(\mathcal{E}_p > \epsilon)$. We use $\epsilon = 4$ for improved stability on RAFT-computed optical flows. We present a visual demonstration of different levels of stereo effects in the supplementary.

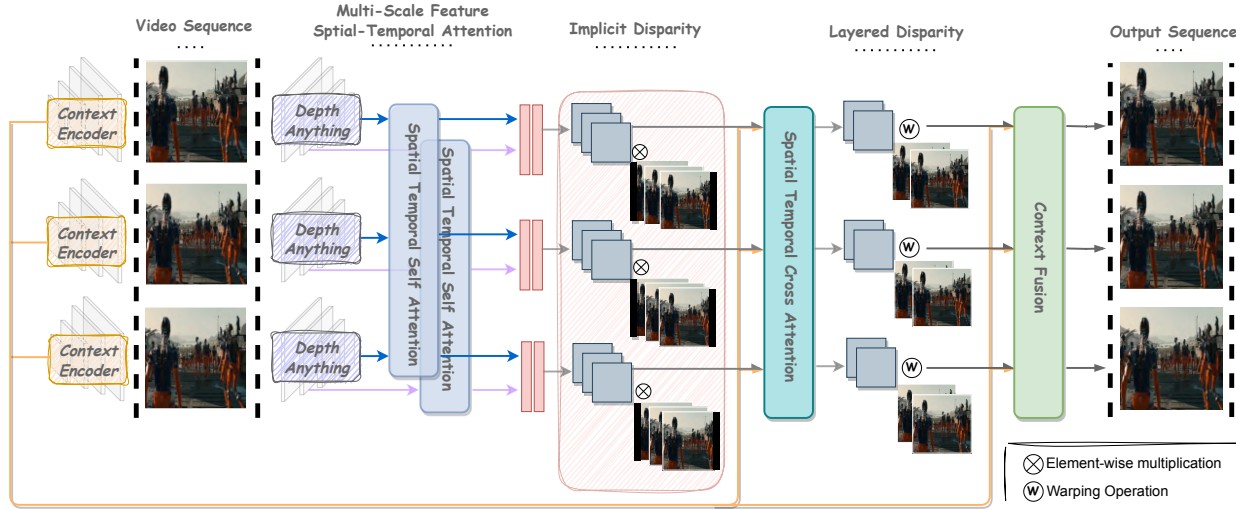

*Figure 2.* Illustration of ImmersePro framework. Our network contains six parts: (1) dual-branch feature extractors for extracting disparity features and context features (Section 4.1), (2) multi-scale spatial-temporal self-attention to refine disparity features (Section 4.2), (3) implicit disparity to generate stereo images without explicit disparities (Section 4.3), (4) spatial-temporal cross attention block to inject contextual information into the implicit disparity features (Section 4.2), (5) layered disparity to obtain the estimated right view video sequences (Section 4.4), and (6) context fusion to enrich the estimated right view video with detailed semantic information (Section 4.5).

*Table 2.* Flow-based consistency check results. Most frames present subtle stereo effects in the dataset.

| occluded area | < 10% | < 20% | < 30% | < 40% |
|---|---|---|---|---|
| Percentage | 71.27% | 84.60% | 91.30% | 94.71% |

## 4. Method

A stereo video sequence $I = \{I^l, I^r\}$ contains left and right video sequences of $I^l \in R^{T \times H \times W \times 3}$ and $I^r \in R^{T \times H \times W \times 3}$, respectively. We use $T, H, W$ to denote the video sequence length, video height, and video width, respectively. We aim to predict a right video sequence $\hat{I}^r$ based on the input left video sequence $I^l$ to make $\hat{I} = \{I^l, \hat{I}^r\}$ presents similar stereo effects as $I$.

As shown in Figure 2, our method comprises six stages. First, we use a dual branch architecture (Section 4.1) that consists of a disparity branch and a context branch, to extract disparity and semantic features, respectively. Second, we apply spatial-temporal self-attention (Section 4.2) on each scale feature to achieve multi-frame awareness. Third, we fuse the multi-scale features to obtain implicit disparity features (Section 4.3). Fourth, we then use a spatial-temporal cross-attention module (Section 4.2) to inject contextual information into the implicit disparity features to obtain layered disparity features (Section 4.4). Fifth, right-view video sequences can be estimated by warping through layered disparities. Finally, we enrich the estimated right-view sequences with a context fusion module.

### 4.1. Dual Branch Architecture

We use a dual-branch architecture to enhance stereo video conversion by separately processing disparity and contextual information, as shown in Figure 2. We employ a pre-trained DepthAnything (Yang et al., 2024) model for the disparity branch to extract disparity-oriented feature maps, while a context feature extractor with the same architecture from (Zhou et al., 2023; Li et al., 2022)'s encoder is used to extract contextual semantic features.

The disparity branch operates on multiple scales, extracting features at $1/2$ and $1/4$ resolutions of the original input to capture detailed disparity information. The disparity features are from the decoder of the model[1]. This branch utilizes spatial-temporal self-attention modules (Section 4.2) to prioritize relevant spatial and temporal details on different scales, ensuring that the model focuses on areas with significant disparity changes or movement. By combining the multi-scale features into $1/2$ resolution with a fusion block, we apply softmax to create a probability distribution that represents the implicit disparities. The implicit disparity is used to select the appropriate pixels from a stack of multiple horizontally shifted copies of the input image (Section 4.3). By encouraging accurate selection, these features implicitly represent the disparity for stereo conversion.

Concurrently, we use a stack of convolution layers as the context encoder. We experimented with multiple encoder

---

[1]We use the output from the neck of the model, as implemented by https://github.com/huggingface/transformers.

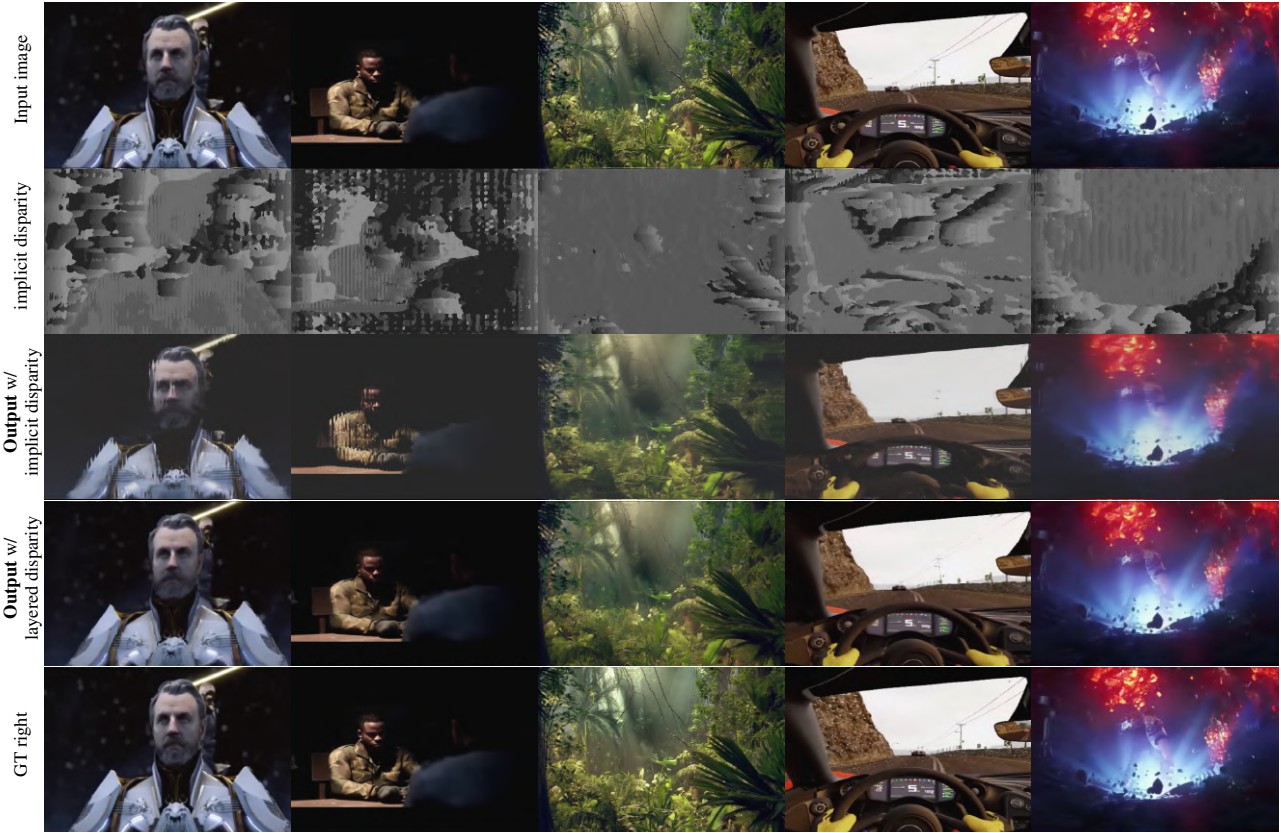

*Figure 3.* Visual demonstration of the implicit disparity guidance. We can observe that *(1)* the implicit disparity module tries to resolve the disparity from the given image, and *(2)* our method can significantly rectify the error introduced by the implicit disparity estimations. Our method offers a significant improvement in clarity with less irregular texture deformation in the image. The implicit disparity map contains multiple channels, and we apply $argmax$ to obtain the visual output.

architectures and settled on the architecture without aggressive downsampling. The details for the context encoder are presented in Appendix A.2. The context branch focuses solely on capturing texture information. This branch processes texture at $1/2$ the original resolution, aligning with the disparity branch's output. Finally, with spatial-temporal cross-attention modules to fuse the implicit disparity and texture information, we apply a layered disparity warping (Section 4.4) to obtain the final predicted right-view.

### 4.2. Spatial-Temporal Attention

Video transformers have demonstrated excellent performances in video segmentation (Duke et al., 2021), video-text feature mapping (Li et al., 2023), and video inpainting (Li et al., 2022; Zhou et al., 2023). This work builds sparse video transformers on top of the ProPainter's version, considering the highly redundant and repetitive textures in adjacent frames. We remove the mask guidance in the original model and use a temporal stride of 2 to avoid redundant key/value tokens within each transformer block and to improve the computational efficiency. Aside from spatial-temporal self-attention, we also use spatial-temporal cross-attention

to fuse features from different sources.

Given a video feature sequence $E_s \in \mathbb{R}^{T_s \times H_s \times W_s \times C}$, we first perform soft split (Liu et al., 2021) to generate patch embeddings $Z \in \mathbb{R}^{T_s \times M \times N \times C_z}$. Subsequently, $Z$ is partitioned into $m \times n$ non-overlapping windows, yielding the partitioned embedding features $Z_w \in \mathbb{R}^{T_s \times m \times n \times h \times w \times C_z}$, where $m \times n$ denotes the number of windows and $h \times w$ denotes their size. For self-attention transformer blocks, we obtain the query $Q$, key $K$, and value $V$ from $Z_w$ through three linear layers, respectively. For cross-attention transformer blocks, we repeat the above process to obtain embeddings $Z_c \in \mathbb{R}^{T_s \times m \times n \times h \times w \times C_z}$ from another feature sequence $E_c \in \mathbb{R}^{T_s \times H_s \times W_s \times C}$. Note that $Z_c$ shares the same shape as $Z_w$. Then $Q$ is extracted from $Z_w$ whilst $K$ and $V$ are extracted from $Z_c$. For both self- and cross-attention, the final embedding features are gathered using soft composition (Liu et al., 2021) for further processing.

### 4.3. Implicit Disparity

For stereo vision, different from common generative tasks, the generated right view requires a precise match to the input

view with as little improvisation as possible. The stereo pair of an image is commonly constructed by obtaining the disparity map to find the shifting distances of each pixel within the input view. Assuming $d_{i,j}$ is the disparity value at pixel location $(i, j)$ in the left image, the corresponding pixel in the right image is:

$$I_{i,j}^r = I_{i,j+d_{i,j}}^l. \tag{2}$$

It is typically a non-differentiable operation due to its piecewise nature. (Jaderberg et al., 2015) propose to use subgradients for backpropagation through spatial transformations to handle such non-smooth operations, enabling differentiable warping.

Another approach (Xie et al., 2016) uses a depth selection layer to align the output right view to the source input view structure. Subsequent works such as (Zhang et al., 2019) follow a similar idea. We employ it as auxiliary supervision. We found this method to be suitable for guidance only. Directly using it to compute the output leads to blurry results. *Implicit disparity* predicts a probability distribution across possible disparity values $d$ at each pixel location. $p_{i,j}^d$, with $\sum_d p_{i,j}^d = 1$, denotes the probability of pixel $(i, j)$ having disparity $d$. We denote an image that is shifted by $d$ pixels horizontally as $I_{i,j}^d = I_{i,j-d}$. We obtain the right view by:

$$\hat{I}_{i,j}^{aux} \sum_d = I_{i,j}^d p_{i,j}^d. \tag{3}$$

where $\hat{I}^{aux}$ is the auxiliary predicted right view. We use $V_{i,j}^d = I_{i,j}^d D_{i,j}^d$ for subsequent computations. This approach estimates the stereo pair of a given image without an explicit disparity map, serving as a relaxation of the warping operation in Equation (2). Without implicit disparity, our model can hardly converge, as shown in Section 5.1.

---

**Algorithm 1** Synthesis from layered disparities.

```
1: # number_layered_disparity:  the number of disparity
   layers.
2: # warped_output:  'BDTCHW'.  A stack of images
   warped by layered disparities.  D is the number of
   disparity layers.
3: # warped_mask:  'BDTCHW'.  A stack of masks warped
   by layered disparities.  D is the number of disparity
   layers.
4: layered_mask = zeros_like(output_mask)
5: total_mask = zeros_like(output_mask)
6: for i in range(number_layered_disparity) do
7:   if i == 0 then
8:     layered_mask[:, i] = warped_mask[:, i]
9:     total_mask[:, i] = warped_mask[:, i]
10:   else
11:     total_mask[:, i] = logical_or(warped_mask[:, i],
       layered_mask[:, i - 1])
12:     layered_mask[:, i] = torch.logical_and((1 -
       total_mask[:, i]), warped_mask[:, i - 1])
13:   end if
14: end for
15: output = layered_mask * warped_output
```

## 4.4. Layered Disparity

The *implicit disparity* is a summation-based approach that computes pixel colors as a blend of other pixel colors, weighted by the estimated probabilities. This may produce good results with a correct estimation, but it may introduce artifacts such as blurring if the estimation is inaccurate. The final output visually improves if each pixel location is selected from a set of candidate disparity layers, rather than blending all the layers. The proposed *Layered Disparity* uses a smaller stack of candidate layers, and each layer represents disparity information. Therefore, our layered disparity representation is a stack of disparity maps. We use a differentiable warping (Jaderberg et al., 2015) operation to warp the input image to an output image. While a single disparity map already defines a solution to the problem, there may be problems due to occlusion and disocclusion artifacts. Our approach avoids the mentioned blending problem. Meanwhile, we maximize the reuse of pixel information within the image while avoiding generating image holes.

We use the implicit disparity $V_{i,j}^d$ as guidance to generate layered disparities. First, we employ three *Conv-ReLU* blocks to refine the $V_{i,j}^d$ to shrink them from $\mathbb{R}^{T_s \times H \times W \times D}$ into $\mathbb{R}^{T_s \times \frac{H}{2} \times \frac{W}{2} \times D}$, where $D$ is the number of stacked disparities. We then apply the spatial-temporal cross-attention process, as mentioned in Section 4.2. With the attention-applied features, a deconvolution operation and three *Conv-ReLU* blocks are used to obtain the final layered disparity $LD_{i,j}^d$. Here, $d = 7$ since we use 7 disparity layers in our work. We then apply the differentiable warping operation with the layered disparity to obtain layered warped images $\hat{I}_{i,j}$ and masks $\hat{M}_{i,j}$, respectively. We select pixel values according to the layered masks as in Algorithm 1. As shown in Figure 3, our proposed approach significantly improved the visual quality compared to the output from the implicit disparity layer. Figure 4 visualizes an example of learned disparity maps from the layered disparity representation.

## 4.5. Context Fusion

The final stage of our network focuses on enriching semantic details while maintaining the semantic structure. The context fusion module integrates semantic and disparity features from a video sequence by concatenating the encoder feature map with layered disparity features to form a fused representation. These fused features are then processed through spatial-temporal attention (Section 4.2), enabling global context awareness. We apply spatial-temporal attention modules at 1/2 the original resolution, as mentioned in Section 4.1. To retain structural integrity, a residual connection reintroduces the refined transformer output into the original fused feature map. We then apply a deconvolution to obtain a texture map in the original resolution, then enrich the texture map by three *Conv-ReLU* blocks. Next,

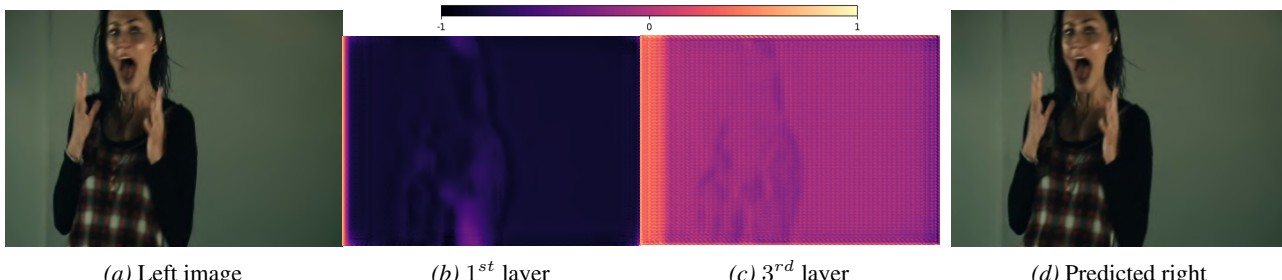

*(a)* Left image      *(b)* $1^{st}$ layer      *(c)* $3^{rd}$ layer      *(d)* Predicted right

*Figure 4.* Visual demonstration of our layered disparity representation. We show the $1^{st}$ and $3^{rd}$ disparity layers in Figures 4b and 4c. We denote darker colors as moving to the right and lighter colors as moving to the left. We use 7 layers in total, while we found the $1^{st}$ and $3^{rd}$ layers contribute to the right-view generation the most. We observe that the $1^{st}$ layer aims to warp the majority of the pixels to the right to their correct right-view location while the $3^{rd}$ layer moves pixels to the left to fill the resulting holes, e.g., near the left border.

the module supplements the layered disparity-warped images from Section 4.4 with the enriched feature map. To be specific, a median blur with $3 \times 3$ kernels is first applied to the warped images to reduce noise and improve local smoothness before concatenating them with the enriched feature map. A semantic residual is then derived by passing this combined map through three *Conv-ReLU* blocks. The final output combines the blurred image with the semantic residual. Therefore, the final result maintains sharp textures while preserving structural consistency, achieving a balance between local detail and global coherence.

## 5. Results

We implement our method using Pytorch and train on four NVIDIA A100 (80G) GPUs for 50,000 iterations (approx. 3 days). Models are trained for 40,000 iterations for our ablations. At training time, we first resize the input sequence to $422 \times 422$ and then randomly crop the resized video sequence to $384 \times 384$. Each input sequence contains 8 frames. We use L1 loss during training to encourage an accurate reconstruction of the right-view images using both implicit and layered disparities. In addition, an LPIPS (Zhang et al., 2018) loss is used for better reconstruction results. An AdamW (Loshchilov & Hutter, 2019) optimizer is used. We use $3e-5$ learning rate while image losses are computed within the range of $(-127.5, 127.5)$. We evaluated our method on our test set, which includes 43 video sequences with 558K frames.

### 5.1. Comparison with State-of-art models

**Benchmark settings.** We compare our method with three state-of-the-art methods including Stereo-from-mono (Watson et al., 2020), 3D Photography (Shih et al., 2020), StereoDiffusion (Wang et al., 2024), and SVG (Dai et al., 2025). Due to the high runtime of those methods (especially for StereoDiffusion which is required to perform inversion (Mokady et al., 2023) for each image), we compare

*Table 3.* Benchmark results. The best and second-best results are highlighted in green and yellow, respectively. * indicates the model is not converged.

|  | L1 ↓ | SSIM ↑ | PSNR ↑ | MEt3R ↓ |
|---|---|---|---|---|
| Deep3D | 0.2215 | 0.1935 | 11.90 | 45.24 |
| 3D Photo | 0.1069 | 0.3463 | 16.36 | 11.97 |
| Stereo Diffusion | 0.0816 | 0.4651 | 18.66 | 9.52 |
| Stereo-from-mono | 0.0646 | 0.5685 | 20.77 | 6.90 |
| SVG | 0.0723 | 0.5865 | 19.69 | 8.19 |
| Ours w/o implicit disparity * | n/a | n/a | n/a | n/a |
| Ours w/o layered disparity | 0.0885 | 0.4717 | 19.05 | 26.72 |
| Ours w/o attention blocks | 0.0593 | 0.5894 | 21.41 | 5.91 |
| Ours w/o context fusion | 0.0588 | 0.5959 | 21.66 | 5.34 |
| Ours | 0.0570 | 0.6048 | 21.83 | 5.29 |

those methods with a subsampled dataset every 3 seconds (72 frames). At test time, we process 8 frames as input where the last 2 frames are taken as reference frames. We use widely employed L1, SSIM, and PSNR to evaluate the quality of the generated stereo pairs.

**Benchmark results.** Our qualitative and quantitative results are shown in Figure 1 and Table 3, respectively. The visual results show that other methods tend to generate right-view images with texture deformations. To be specific, 3D photo struggles to find accurate depth cues, resulting in inaccurate warping. Stereo-from-mono can generate images well, but often comes with unpleasant black dots around the warping shapes. StereoDiffusion uses null-text inversion (Mokady et al., 2023) to convert an image to the latent space, then warp the latent features to create the right-view image. It highly depends on the performance of the inversion, resulting in unstable performances. As shown in Table 3, our method yields better numerical results. This finding aligns with the visual results.

Our method achieves the best temporal consistency with a higher inference speed (evaluated on a single NVIDIA A100 GPU), as shown in Table 4. We use the default input resolution for each method, and note that image-based methods are applied frame-by-frame, while video-based methods are

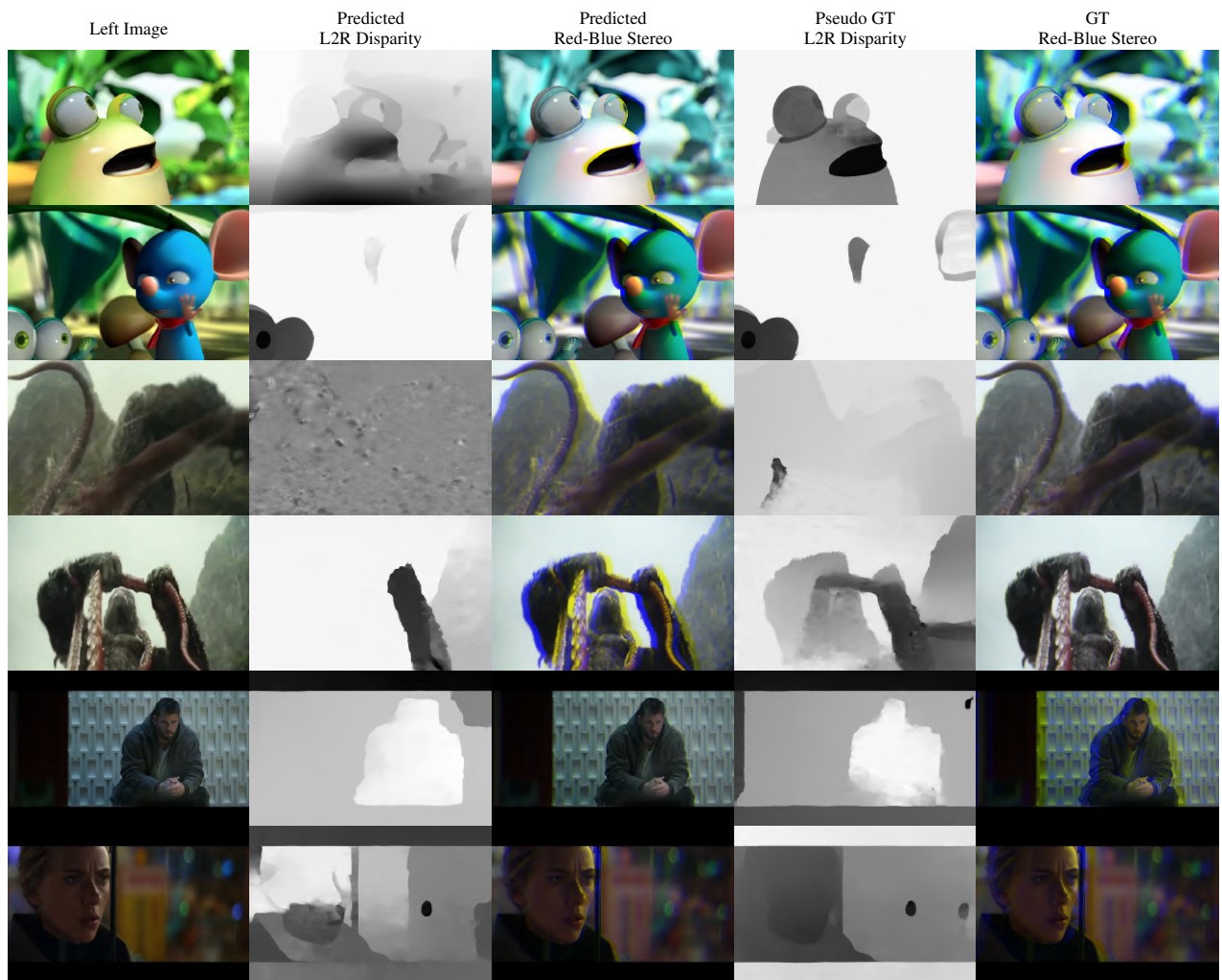

*Figure 5.* Visual demonstration of the disparity analysis results. Our network predicts reasonable stereo effects but may be stronger or weaker if compared to the ground truth. The L2R disparity computes the left-to-right disparity using RAFT-Stereo (Lipson et al., 2021).

applied clip-by-clip. In addition, our accompanying videos demonstrate better stability in terms of jittering and shaking. Please watch the accompanying videos at 0.5 speed to see the artifacts generated by the different methods.

*Table 4.* Other performance results.

|  | 3D Photo | Stereo Diffusion | Stereo-from-mono | ours |
|---|---|---|---|---|
| FVD ↓ | 35.04 | 18.97 | 3.71 | **2.88** |
| Inference Speed (FPS) ↑ | 0.004 | 0.137 | 1.197 | **5.609** |

**Ablation results.** Table 3 shows our ablation results. We show that our method is not going to converge without using implicit disparity guidance, while a significant performance drop may occur when removing our proposed layered disparity. We show that our layered disparity generates better visual quality in Figure 3 compared to the outputs from implicit disparities. Though not significant, the attention blocks can slightly improve the overall performance, while

the context fusion module contributes significantly. Additional experiments, including alternative masking strategies, the inclusion of the context fusion module, flow-guided feature propagation, and different backbone choices, are included in our supplementary material. Lastly, we show our method may generate different levels of stereo effects in Figure 5 compared to the ground truth, but this is expected due to the underdetermined nature of the problem, and we consider our solution also as reasonable.

**User Study.** We present a user study with 26 participants (computer vision researchers), using a Meta Quest 3 headset. The study assessed four key dimensions on a 5-point Likert scale (higher is better):

- *Frame Quality*: Evaluated using generated right-eye frames only. Watched as 2D videos.
- *Temporal Coherence*: Evaluated via 3D stereo playback.

- *Stereoscopic Effects*: The strength of stereo effects.
- *Overall Conformity*: General visual experience from the user's perspective.

The users will first take a look at the generated right frames only for the frame quality measure, then play the videos in 3D left-right stereo mode for the other measures.

*Table 5.* User study results comparing preference scores (1–5). The highest and second-highest values are highlighted in **red** and **cyan**, respectively.

| | 3D Photo | StereoDiff. | Stereo-from-mono | SVG | Ours |
|---|---|---|---|---|---|
| Frame Quality | 2.50 | 2.77 | **3.00** | 2.76 | **4.12** |
| Temporal Coherence | 2.35 | 2.31 | **3.17** | 2.94 | **4.23** |
| Stereoscopic Effects | 2.38 | 2.61 | 3.00 | **3.04** | **3.92** |
| Overall Conformity | 2.38 | 2.57 | **3.04** | 2.88 | **4.23** |

As shown in Table 5, our method outperforms existing baselines across all categories. Notably, the gain in Temporal Coherence and Overall Conformity suggests that our approach provides a more stable and immersive viewing experience in a VR environment.

## 6. Discussion And Limitation

To enhance the viewing experience, films sometimes employ a stronger stereoscopic effect at the start and end, while moderating it in the middle to ensure viewer comfort (Neuman, 2009; Ranftl et al., 2022). Thus, the stereo parameters, such as focal length, are hard to retrieve even for the same film. Theoretically, the precise reproduction of the right view is impossible without knowing the stereo parameters in advance. By learning through a large-scale dataset, *ImmersePro* estimates its average disparity, then tries to create an average-level stereo effect for input videos rather than reproduce the precise right pair. Therefore, as shown in Figure 6, our model may produce reasonable but "inaccurate" stereo effects if compared with the ground truth.

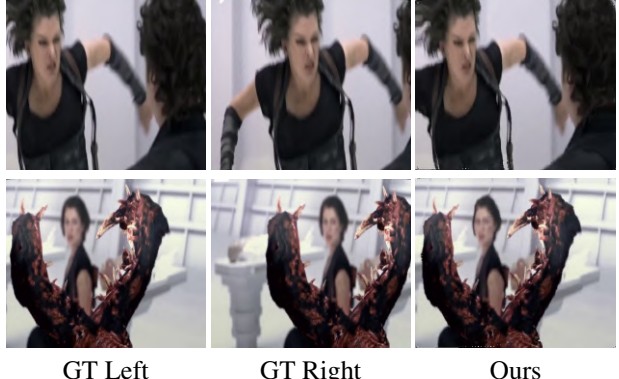

GT Left      GT Right      Ours

*Figure 6.* Challenging examples with unusually strong stereo effects. While the ground-truth stereo pair exhibits large disparities, our method produces moderate stereo effects.

In addition, our framework enforces temporal consistency within a local sliding window, generating videos in short clips with overlapping reference frames. While this strategy yields good results in practice, it may limit consistency over longer temporal horizons. Future work could explore incorporating global or rolling alignment mechanisms, such as approaches inspired by RollingDepth (Ke et al., 2025), to further improve long-range temporal coherence.

A reasonable stereo conversion pipeline involves a warping-and-inpainting process, where the inpainting operation fills the black holes created by the warping operation. One sample work is stereo-from-mono (Watson et al., 2020) that performs inpainting with a randomly sampled image from the training dataset. In a way, our method can be seen as an improvement to stereo-from-mono by intelligently selecting the correct regions for inpainting. However, this strategy works for creating stereo movies with "subtle" stereo effects without the need for significant inpainting. As we observed in most 3D movie examples, very few movies contain strong stereo effects. Notably, our method cannot produce strong stereo effects due to the limited dataset and limited inpainting capabilities. In future work, we would like to investigate how Nerf (Mildenhall et al., 2021)-based inpainting can be used for stereo-movie generation.

## 7. Conclusion

This work presents an end-to-end video-based stereo conversion method that generates right-view video sequences according to the input video. Our method automatically utilizes layered disparity maps on top of implicit disparities. Additionally, we propose *Youtube-SBS*, a large-scale stereo dataset that is publicly available for benchmarking purposes. Extensive qualitative and quantitative evaluations demonstrated the robustness of our approach against previous works.

## Acknowledgments

This work is supported by funding from King Abdullah University of Science and Technology (KAUST) - Center of Excellence for Generative AI, under award number 5940 and a gift from Google.

## Impact Statement

This work presents a fast and scalable method for stereo video conversion, enabling more accessible creation of immersive 3D content for applications such as virtual reality, filmmaking, and digital media.

Potential risks arise from its application to synthetic or manipulated videos, where improved realism could amplify misleading content. These concerns are not unique to our

method but are shared across many video processing and generative techniques. Responsible use and clear disclosure of synthesized content are important to mitigate such risks.

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

# Appendix

We present implementation details and additional experiments in our supplementary material. Please watch the accompanying videos at 0.5 speed to see the artifacts generated by the different methods.

## A. Technical Details

In this section, we provide an evaluation of our dataset and an illustration of the context encoder employed in our framework.

### A.1. Dataset Evaluation

We present demographic data and a detailed visualization of stereoscopic effects in this section to provide a more intuitive understanding of our proposed dataset.

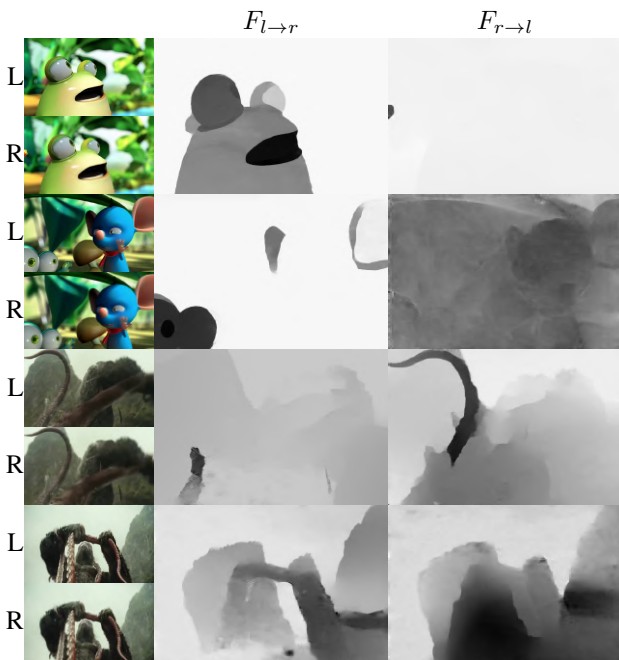

*Figure 7.* Visual demonstration of optical flow estimation with *RAFT-Stereo*. The left-to-right estimation is reasonable while right-to-left estimation tends to fail.

#### A.1.1. VISUAL REFERENCE FOR STEREOEFFECTS

We provide a visual reference for the *RAFT*-based optical flow analysis of the proposed *Youtube-SBS* dataset in Figure 9.

**Data Cleaning**   We use *RAFT* to screen the videos, then manually verify if a video or a frame should be excluded or not. Thus, we denote that the videos used are not algorithmically biased. In general, the filtered videos are of two types: 1) videos that are not SBS but claimed as SBS stereo in their

titles on YouTube, and 2) corrupted SBS videos with inconsistent left and right frames. We show two representative cases as follows:

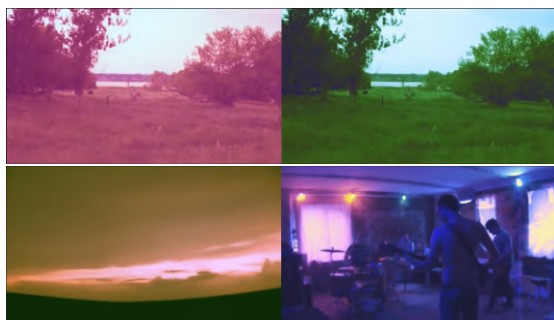

*Figure 8.* Internet SBS videos with apparent errors.

**Optical Flow Estimation Method**   As we stated in our main text, we found the current optical flow estimation methods for stereo pairs can perform good for left-to-right but not right-to-left estimation, while a generic estimator such as RAFT can present an overall stable performance. We present visual evidence for the failed cases in Figure 7.

### A.2. Context Encoder

Our context encoder uses a stack of convolutional layers to obtain semantic features from the input image as below. Starting from the $5^{th}$ convolution layer, the extracted features are the concatenation of the features from the current and previous layers.

```
Conv2d(3, 64, kernel_size=3, stride=1, padding=1),
LeakyReLU(0.2, inplace=True),
Conv2d(64, 64, kernel_size=3, stride=2, padding=1),
LeakyReLU(0.2, inplace=True),
Conv2d(64, 128, kernel_size=3, stride=1, padding=1),
LeakyReLU(0.2, inplace=True),
Conv2d(128, 256, kernel_size=3, stride=1, padding=1),
LeakyReLU(0.2, inplace=True),
Conv2d(256,384,kernel_size=3,stride=1,padding=1,groups=1),
LeakyReLU(0.2, inplace=True),
Conv2d(640,512,kernel_size=3,stride=1,padding=1,groups=2),
LeakyReLU(0.2, inplace=True),
Conv2d(768,384,kernel_size=3,stride=1,padding=1,groups=4),
LeakyReLU(0.2, inplace=True),
Conv2d(640,256,kernel_size=3,stride=1,padding=1,groups=8),
LeakyReLU(0.2, inplace=True),
```

## B. Additional Experiments

### B.1. Flow-Guided Feature Propagation

Video feature propagation and deformation have shown their effectiveness for many video-based tasks (Xue et al., 2019; Wang et al., 2019b; Haris et al., 2019). The flow-guided deformation concept is particularly suitable for the stereo conversion scenario as the pixel shifting nature according to the disparities. Similar to E$^2$FGVI (Li et al., 2022) and ProPainter (Zhou et al., 2023), we use a similar design of flow-guided feature propagation module, that features bidirectional optical flow-guided deformable alignments that

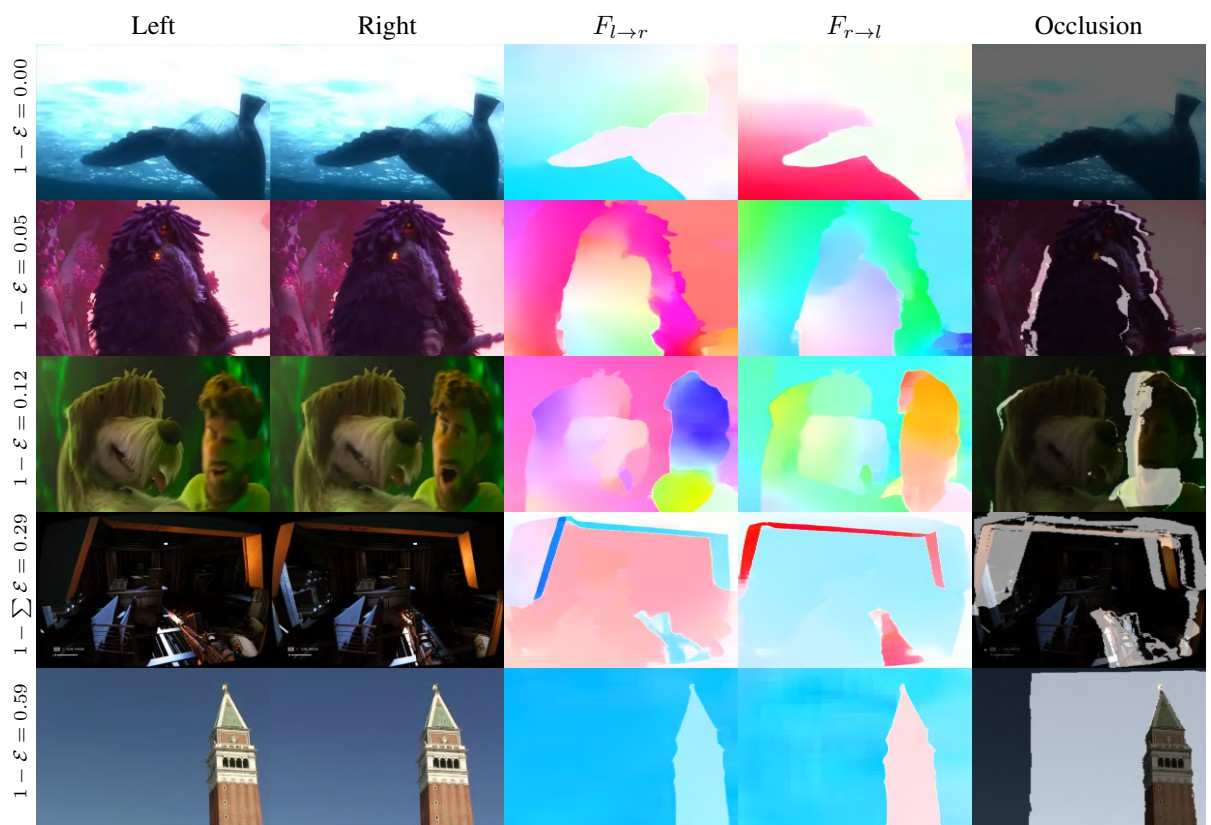

*Figure 9.* Visual demonstration of the different levels of stereo effects with the dataset.

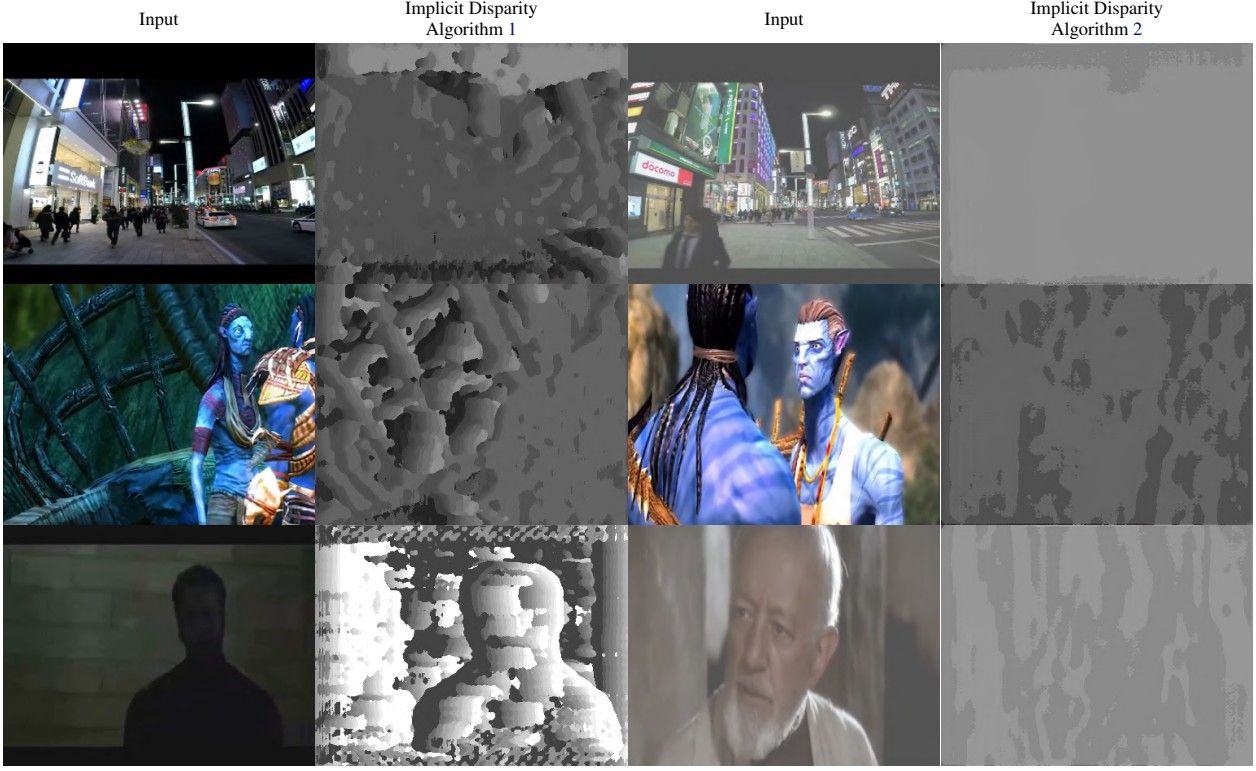

*Figure 10.* Visual demonstration of the implicit disparity output for different masking strategies. The implicit disparity map contains multiple channels and we apply $argmax$ to obtain the visual output.

built on top of deformable convolution networks (DCN) (Dai et al., 2017; Zhu et al., 2019).

Given extracted features $\{E_t | t = 1...T\}$ from a feature encoder where $T$ is the total number of frames. Under the context of stereo conversion, the forward flow $F_{t \to t+1}$ helps to track the movement of occluded regions from frame $t$ to frame $t + 1$. When the pixels within the occluded areas of frame $t$ are found in the valid regions of frame $t + 1$, this information can be utilized effectively by warping the backward propagation feature $\hat{E}_b^{t+1}$ from frame $t + 1$ back to frame $t$, guided by the forward flow $F_{t \to t+1}$. On top of $E^2$FGVI's approach, we include flow validation maps $M_{t+1 \to t}$ by consistency check introduced by ProPainter. The consistency check compares the forward and backward optical flows to ensure the correctness of the used optical flows. Similar to Equation (1), the consistency error is computed as follows:

$$\mathcal{E}_{t \to t+1}(p) = ||F_{t \to t+1}(p)) + F_{t+1 \to t}(p + F_{t \to t+1}(p))||_2^2, \quad (4)$$

where $p$ is pixel positions of the frame. Then the flow deformation offsets $\tilde{o}_{t \to t+1}$) are computed with the DCN network, where a concatenation of the forward flow $F_{t \to t+1}$, backward propagation feature $\hat{E}_b^{t+1}$, warped backward feature $\mathcal{W}(\hat{E}_b^{t+1}, F_{t \to t+1})$, and flow validation maps $M_{t+1 \to t}$ is used as the condition, where $\mathcal{W}$ is warping operation. The flow-guided alignment propagation is then:

$$\hat{E}_b^t = \mathcal{R}(\mathcal{D}(\hat{E}_b^{t+1}; F_{t \to t+1} + \tilde{o}_{t \to t+1}), f_t), \quad (5)$$

where $\mathcal{D}(\cdot)$ is the deformable convolution layers and $\mathcal{R}(\cdot)$ fuses the aligned and current features.

However, we found the disparity cannot be learned with those flow-guided propagation modules. We suspect the feature map deformation and alignment can break the internal disparity features, resulting in a failed learning of the implicit disparity maps.

### B.2. Different Backbones For the Depth Branch

We provide additional results in Table 6. We experimented with MiDaS instead of DepthAnything. The results indicate that different depth estimation backbones do not affect the performance of our proposed method.

| | Depth Backbone | L1 ↓ | SSIM ↑ | PSNR ↑ |
|---|---|---|---|---|
| Ours w/o context fusion | MiDaS | 0.0590 | 0.6014 | 21.6572 |
| Ours w/o context fusion | DepthAnything | 0.0588 | 0.5959 | 21.6649 |

*Table 6.* Additional results. The best results are highlighted in green.

### B.3. Alternative Mask Selection Algorithm

To avoid having multiple pixels being mapped to the same pixel location $i, j$, we use an algorithm to produce $[0, 1]$

masks so that different layers cannot interfere with each other, as shown in Algorithm 1. In addition, we further tested another design where the mask value selection algorithm Algorithm 2 generates mask values $\in \{-1, 0, 1\}$, to allow more interactions between layers. However, though Algorithm 2 can better resolve complicated scenarios, we found the intermediate implicit disparity layers often fail to resolve disparities correctly, as shown in Figure 10. In general, we found that the Algorithm 2 tries to weaken the disparity cues, resulting in smoother output with weaker or wrong disparity maps.

---

**Algorithm 2** Synthesis from layered disparities.

```
1: # number_layered_disparity:  the number of disparity
   layers.
2: # warped_output: 'BDTCHW'.  A stack of images
   warped by layered disparities.  D is the number of
   disparity layers.
3: # warped_mask: 'BDTCHW'.  A stack of masks warped
   by layered disparities.  D is the number of disparity
   layers.
4: layered_mask = zeros_like(output_mask)
5: total_mask = zeros_like(output_mask)
6: for i in range(number_layered_disparity) do
7:   if i == 0 then
8:     layered_mask[:, i] = warped_mask[:, i]
9:     total_mask[:, i] = warped_mask[:, i]
10:   else
11:     total_mask[:, i] = logical_or(warped_mask[:, i],
      layered_mask[:, i - 1])
12:     layered_mask[:, :, i] = total_mask[:, :, i] -
      output_mask[:, :, i - 1]
13:   end if
14: end for
15: output = layered_mask * warped_output
```

---

### B.4. Context Fusion Module

As shown in Table 3, the inclusion of the context fusion module significantly enhances the overall statistical performance. Moreover, as demonstrated in the accompanying videos, this module greatly improves the temporal consistency of the generated videos. However, we observed potential artifacts in frames with complex feature patterns, as illustrated in Figure 11. We suspect that these edge cases could be mitigated with a larger training dataset that includes greater internal variance, allowing the model to better handle such intricate scenarios.

| GT Left | w Context Fusion | w/o Context Fusion |

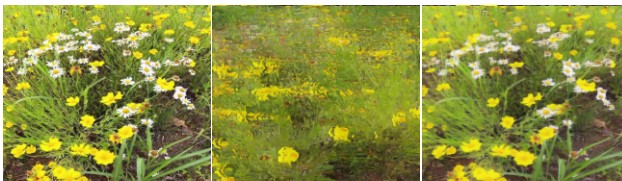

*Figure 11.* Visual demonstration of the failed edge cases of context fusion.

