# OpenReview forum: "ImmersePro: End-to-End Stereo Video Synthesis Via Implicit Disparity Learning"
_ICML.cc/2026/Conference — ICML 2026 regular_

### Official Review · Reviewer_RNMS · 2026-02-24

**Soundness:** 3
**Presentation:** 3
**Significance:** 3
**Originality:** 3
**Overall Recommendation:** 4
**Confidence:** 5

**Summary:**

The paper introduces "ImmersePro," an end-to-end framework designed to convert monocular (2D) videos into stereoscopic (3D) videos. Unlike previous image-based methods that often suffer from temporal instability or artifacts, this approach utilizes a dual-branch architecture (disparity and context branches) enhanced with spatial-temporal attention mechanisms. A key technical contribution is the use of "Implicit Disparity" guidance—predicting a probability distribution of disparities rather than a single explicit map—followed by a "Layered Disparity" module to handle occlusions and generate the right-eye view via differentiable warping. Additionally, the authors release "YouTube-SBS," a dataset of 423 stereo videos (over 7 million pairs) curated from YouTube to serve as a benchmark. Experiments show improvements in L1, SSIM, and PSNR metrics compared to state-of-the-art baselines like StereoDiffusion and 3D Photo.

**Compliance With Llm Reviewing Policy:**

Affirmed.

**Key Questions For Authors:**

- Evaluation of Stereoscopic Quality: Given your acknowledgment that the ground truth stereo parameters are unknown and the model learns an "average," do pixel-wise metrics (L1/PSNR) accurately reflect the quality of the 3D effect? Did you consider using a perceptual metric specific to stereo vision or conducting a user study to evaluate depth comfort?

- Dataset Bias: By filtering the YouTube-SBS dataset based on RAFT optical flow consistency (removing frames with high occlusion/disocclusion errors), are you inadvertently training and testing on "easy" samples? How does the model perform on scenes with high occlusion which were excluded from the dataset construction?

- Generalization: How does the model handle scenes that deviate significantly from the "average disparity" learned from the dataset (e.g., a scene shot with an unusually wide baseline)?

**Limitations:**

The authors candidly discuss limitations in Section 6. They note that the model cannot reproduce precise ground truth stereo parameters, instead opting for a learned "average" effect. They also mention the inability to produce "strong" stereo effects due to the dataset distribution and inpainting limitations. Another unstated limitation is the dependence on the domain of the training data; movie trailers often have quick cuts, which might impact the temporal attention mechanism's ability to learn long-term consistency compared to continuous shots.

**Strengths And Weaknesses:**

Strengths:

- Novel Methodology: The shift from explicit disparity maps to "implicit disparity" (probability distributions) combined with "layered disparity" is a clever approach to handle the inherent ambiguity and occlusion issues in 2D-to-3D conversion.

- Temporal Consistency: The integration of spatial-temporal self-attention and cross-attention addresses a major pain point in video synthesis—flicker and temporal inconsistency—which single-image baselines often fail to resolve.

- Dataset Contribution: The creation and release of "YouTube-SBS" is a significant contribution. Existing datasets are often synthetic (Sintel), driving-focused (KITTI), or small. A large-scale, movie-centric dataset fills a gap in the literature.

- Performance: The method quantitatively outperforms baselines in standard metrics (L1, SSIM, PSNR) and offers significantly faster inference speeds than diffusion-based competitors (5.6 FPS vs 0.137 FPS).

Weaknesses:

- Evaluation Metric vs. Problem Nature Contradiction: In the Discussion (Section 6), the authors admit that exact reproduction of the ground truth right view is theoretically impossible due to unknown stereo parameters (focal length, baseline). They state the model learns an "average disparity." However, they rely heavily on pixel-perfect metrics (L1, PSNR) for evaluation. If the model generates a valid but different stereo depth than the ground truth, these metrics will penalize it heavily. This disconnect undermines the reliance on quantitative metrics.

- Lack of Subjective Evaluation: For a visual synthesis task—specifically one aimed at human viewing comfort (3D movies)—the absence of a user study is a notable deficiency. Metrics like SSIM do not fully capture stereoscopic comfort or depth perception quality.

- Dataset Filtering Bias: The dataset curation involved filtering out videos where RAFT optical flow consistency was low (Section 3). While this ensures cleaner data, it potentially biases the dataset toward "easy" scenes with minimal complex occlusions, which might inflate performance results.

- Typographical Errors: There are minor presentation issues, such as "Sptial-Temporal" in Figure 2.

---

> ### Author Rebuttal · Authors · 2026-03-30
>
> 1. **Dataset Bias**: Our dataset contains over 7M frames that cannot be easily filtered manually. We use RAFT to screen the videos, then manually verify if a video or a frame should be excluded or not. Thus, the videos used are **NOT** algorithmically biased. In general, the filtered videos are of two types: 1) videos that are not SBS but claimed as SBS stereo in their titles on YouTube, and 2) corrupted SBS videos with inconsistent left and right frames. We will add further clarification in our revised version, along with representative images to show the filtered incorrect video frames.
>
> 1. **Evaluation of Stereoscopic Quality**: We present pixel-wise metrics to align with prior works, which remain a meaningful proxy for evaluating the consistency with the GT distribution. We agree with the reviewer that those metrics may not be able to accurately reflect the quality of the 3D effect.  Following the suggestion from Reviewer 9xGx, we adopt MEt3R to evaluate the geometric consistency of the generated stereo pairs. As shown in the table below, our method achieves the best 3D structural consistency among all compared methods, outperforming the nearest competitor (Stereo From Mono) by **23.3\%**.
>
>     |                 | 3D Photo | Stereo Diffusion | Stereo From Mono | Ours     |
>     |:----------------|:---------|:-----------------|:-----------------|:---------|
>     |         MEt3R ↓ | 11.97    | 9.52             | 6.90             | **5.29** |
>
> 1. **User study**: We agree that perceptual evaluation is important.  Due to time constraints during the rebuttal period, we present a pilot user study with only 7 participants (PhD students in computer vision), using a Meta Quest 3 headset. The study assessed four key dimensions on a 5-point Likert scale (higher is better):
>
>     - Frame Quality: Evaluated using generated right-eye frames only. Watch as 2D videos.
>     - Temporal Coherence: Evaluated via 3D stereo playback.
>     - Stereoscopic Effects: The strength of stereo effects.
>     - Overall Conformity: General visual experience from the user's perspective.
>
>     The study evaluates on four aspects, including frame quality, temporal coherence, stereoscopic effects, and overall conformity, in a 5-point Likert scale. We use a Meta Quest 3 headset for the evaluation. The users will first take a look at the generated right frames only for the frame quality measure, then play the videos in 3D left-right stereo mode for the other measures.
>
>     |                      | 3D Photo | Stereo Diffusion | Stereo From Mono | Ours     |
>     |:---------------------|:---------|:-----------------|:-----------------|:---------|
>     |  Frame Quality       | 2.57     | 3.57             | 3.28             | **4.28** |
>     | Temporal Coherence   | 2.28     | 3.00             | 3.57             | **4.14** |
>     | Stereoscopic Effects | 2.57     | 3.28             | 3.42             | **3.85** |
>     | Overall Comformity   | 2.14     | 3.28             | 3.57             | **4.14** |
>
>     Our method outperforms existing baselines across all categories. Notably, the gain in Temporal Coherence and Overall Conformity suggests that our approach provides a more stable and immersive viewing experience in a VR environment.
>
>     While this pilot study provides strong initial evidence of our method's superiority, we acknowledge the small participant pool. We will present a comprehensive user study with at least 30 participants in the revised version of the manuscript.
>
> 1. **Scenes with Unusually Wide Baseline**: We thank the reviewer for raising such an interesting question. Our method receives only the input left videos without any stereo configurations, such as baseline distances. Thus, the generated right videos would still have moderate stereo effects, disregarding the wide baseline of the GT right view. We will add a figure to the supplementary material showing the comparison against videos with unusually wide baselines.
>
> 1. **Quick cuts**: We thank the reviewer for this observation. Although movie trailers contain frequent cuts, these transitions are typically longer than our temporal window (e.g., 8 frames), enabling the model to learn consistent temporal patterns within each segment. As a result, the effect on local temporal consistency is limited, though extending to long-term consistency remains an open problem. We will note this in our discussion in the revised version.

---

> > ### Author Rebuttal · Reviewer_RNMS · 2026-04-05
> >
> > No further comments.

---

### Official Review · Reviewer_UogK · 2026-03-11

**Soundness:** 2
**Presentation:** 2
**Significance:** 3
**Originality:** 2
**Overall Recommendation:** 4
**Confidence:** 3

**Summary:**

The paper introduces ImmersePro, an end-to-end framework designed for converting monocular videos into stereoscopic (3D) videos. The core contribution is a dual-branch architecture consisting of a disparity branch and a context branch, which predicts the corresponding right-view frames from a single-view input. By leveraging a spatio-temporal attention mechanism, the model maintains consistency across video frames, addressing the flickering issues common in frame-by-frame conversion methods. A key technical innovation is the use of implicit disparity guidance and multi-layer disparity representation, which avoids the artifacts typically caused by explicit depth estimation and warping errors.

**Compliance With Llm Reviewing Policy:**

Affirmed.

**Key Questions For Authors:**

see weakness

**Limitations:**

see weakness

**Strengths And Weaknesses:**

While the implicit disparity approach improves visual quality, the underlying geometric interpretability is somewhat reduced compared to traditional depth-based methods, which might lead to inaccuracies in scenes with highly complex occlusions or extreme perspective shifts. The temporal consistency is primarily maintained within a local sliding window (e.g., 8 frames), and the paper could benefit from a discussion on how the model handles long-term consistency in extended video sequences.

Additionally, while the quantitative improvements are clear, stereoscopic video quality is highly subjective; the lack of a formal user study to evaluate perceived depth and visual comfort is a notable omission. Lastly, the computational efficiency and inference latency (FPS) are not explicitly detailed, which are critical factors for the practical deployment of video conversion tools.

---

> ### Author Rebuttal · Authors · 2026-03-30
>
> 1. **Temporal Consistency for Longer Videos**: In our current framework, temporal consistency is primarily enforced within a local sliding window, with two reference frames and six resulting frames. Our current strategy essentially generates a video clip six frames by six frames, with the last two frames included in the current batch as the reference frames for the next batch. While we already get good results for temporal consistency in longer videos, even better results could be achieved by additional components in future work. One possible approach is to use methods such as RollingDepth to realign geometric information across extended temporal horizons on a rolling basis. We will add this discussion to the revised version.
>
> 2. **User study**: We agree that perceptual evaluation is important. Due to time constraints during the rebuttal period, we present a pilot user study with only 7 participants (computer vision researchers), using a Meta Quest 3 headset. The study assessed four key dimensions on a 5-point Likert scale (higher is better):
>
>     - Frame Quality: Evaluated using generated right-eye frames only. Watch as 2D videos.
>     - Temporal Coherence: Evaluated via 3D stereo playback.
>     - Stereoscopic Effects: The strength of stereo effects.
>     - Overall Conformity: General visual experience from the user's perspective.
>
>     The study evaluates on four aspects, including frame quality, temporal coherence, stereoscopic effects, and overall conformity, in a 5-point Likert scale. We use a Meta Quest 3 headset for the evaluation. The users will first take a look at the generated right frames only for the frame quality measure, then play the videos in 3D left-right stereo mode for the other measures.
>
>     |                      | 3D Photo | Stereo Diffusion | Stereo From Mono | Ours     |
>     |:---------------------|:---------|:-----------------|:-----------------|:---------|
>     |  Frame Quality       | 2.57     | 3.57             | 3.28             | **4.28** |
>     | Temporal Coherence   | 2.28     | 3.00             | 3.57             | **4.14** |
>     | Stereoscopic Effects | 2.57     | 3.28             | 3.42             | **3.85** |
>     | Overall Comformity   | 2.14     | 3.28             | 3.57             | **4.14** |
>
>     Our method outperforms existing baselines across all categories. Notably, the gain in Temporal Coherence and Overall Conformity suggests that our approach provides a more stable and immersive viewing experience in a VR environment.
>
>     While this pilot study provides strong initial evidence of our method's superiority, we acknowledge the small participant pool. We will present a comprehensive user study with at least 30 participants in the revised version of the manuscript.
>
> 1.  **Computational efficiency and inference FPS**: Results for inference speed (FPS) are reported in Table 4. All methods are evaluated on a single NVIDIA A100 GPU. We note that image-based methods are applied frame-by-frame, while video-based methods are applied clip-by-clip. We use the default input resolution for each method and report end-to-end throughput under these settings. We will clarify these details in the revised version.

---

> > ### Author Rebuttal · Reviewer_UogK · 2026-04-04
> >
> > Thanks for response.

---

### Official Review · Reviewer_9LfB · 2026-03-11

**Soundness:** 3
**Presentation:** 2
**Significance:** 3
**Originality:** 3
**Overall Recommendation:** 4
**Confidence:** 2

**Summary:**

This paper presents ImmersePro, a novel framework for converting monocular videos to stereoscopic (stereo) videos. The core technical contribution is a dual-branch architecture that leverages spatial-temporal attention mechanisms to generate stereo pairs. Instead of relying on explicit disparity maps, the framework uses an implicit disparity guidance strategy, which the authors argue reduces errors propagated from inaccurate disparity estimation. To support training and benchmarking in this domain, the paper also introduces the YouTube-SBS dataset, a large-scale collection of 423 real-world stereo videos comprising over 7 million frames. Experimental results demonstrate that ImmersePro achieves quantitative improvements over existing state-of-the-art methods, specifically outperforming the stereo-from-mono competitor by 11.76% (L1), 6.39% (SSIM), and 5.10% (PSNR).

**Compliance With Llm Reviewing Policy:**

Affirmed.

**Key Questions For Authors:**

(1) Can experimental comparisons be conducted with more related methods to fully demonstrate the superiority of the proposed approach in this paper?

(2) Can the textual description of the method in the paper be revised to make it clearer and more comprehensible (for example, the caption of Figure 2 is almost identical to the text description in Section 4.Method and lacks conciseness)?

(3) Can the technical limitations and social impacts of the proposed method in this paper be thoroughly discussed?

**Limitations:**

The paper discusses the technical limitations of the method, but it does not include a dedicated discussion on potential negative social impacts. It is hoped that the author can candidly address the possible social implications of the method presented in this paper.

**Strengths And Weaknesses:**

Soundness: This paper proposes an innovative framework named ImmersePro and conducts experimental comparisons on the YouTube-SBS dataset introduced in the paper. The experimental results show that compared to stereo-from-mono, ImmersePro achieves quantitative improvements of 11.76% (L1 error), 6.39% (SSIM), and 5.10% (PSNR), respectively. However, the paper compares relatively few methods in its experiments, raising concerns about the persuasiveness of the experimental results.

Presentation: The overall structure of the paper is relatively clear, but some parts are not described clearly enough (for example, the caption of Figure 2 is almost identical to the text description in Section 4.Method and lacks conciseness). Suggestions for revision are provided.

Significance & Originality: This paper proposes an innovative framework named ImmersePro, specifically designed for converting single-view videos into stereo videos. It also contributes the YouTube-SBS dataset. This work holds significant impact and contribution within this research field.

---

> ### Author Rebuttal · Authors · 2026-03-30
>
> 1. **Comparison with more related methods**: We thank the reviewer for the advice. We further evaluate the more recent SVG (ICLR 2025) approach, which achieves mid-tier performance of L1=19.1531, SSIM=0.5804, PSNR=18.27, MEt3R=9.82. For generating a 16-frame video, SVG takes 540 seconds at an FPS of 0.0018. Due to its slowness, we evaluate only the first 160 frames for each testing video (32 videos in total) in the rebuttal. The numbers will be slightly different in our revised version.
>
> 2. We thank the reviewer for the careful reading. We will improve the conciseness of our caption in our revised version.
>
>
> 3. **Discussion of Limitations and Societal Impact**:
>
>     - **Technical Limitations**:
>         One additional technical limitation, besides the limitations described in the paper, is in long-sequence generation. Our model operates in a feed-forward manner, and maintaining strict geometric and visual consistency across long video sequences remains a challenge.
>
>         Also, for complex scenes with significant occlusion or fast-moving foreground objects over long periods, the model may occasionally struggle to maintain the structural integrity of the synthesized view, as it relies on local temporal context rather than a holistic understanding of the entire video sequence.
>
>     - **Social Impacts**:
>         Our work proposes a fast feed-forward and scalable stereo video conversion method, enabling efficient generation of immersive 3D content for applications such as virtual reality, filmmaking, and digital media.
>         Our method does not generate new videos from scratch, and therefore, there is little risk of generating problematic content. However, our method could be applied to problematic content, such as problematic AI-generated videos.
>         Overall, we do not foresee major societal concerns specific to our method. We will elaborate on the statement in our revised version.

---

> > ### Author Rebuttal · Reviewer_9LfB · 2026-04-03
> >
> > The authors have addressed my concern about insufficient baseline comparisons by adding results from the more recent SVG (ICLR 2025) method. They have committed to improving the conciseness of the figure caption in the revised version. Additionally, they provided a thorough discussion of technical limitations (e.g., long-sequence consistency, occlusion, fast-moving objects) and social impacts, while noting that major societal concerns are not anticipated. All my original concerns are adequately resolved.

---

### Official Review · Reviewer_9xGx · 2026-03-11

**Soundness:** 2
**Presentation:** 2
**Significance:** 2
**Originality:** 2
**Overall Recommendation:** 4
**Confidence:** 3

**Summary:**

### Summary of "ImmersePro: End-to-End Stereo Video Synthesis Via Implicit Disparity Learning"

The paper introduces ImmersePro, a framework for converting monocular videos into stereo (3D) videos by predicting plausible right-eye views. The system uses a dual-branch architecture (disparity branch with pretrained DepthAnything + context branch for semantics/texture) and introduces implicit disparity guidance, predicting a probability distribution over pixel shifts as a differentiable relaxation of warping, rather than relying on explicit disparity maps. A layered disparity representation (7 layers) reduces occlusion artifacts. The paper also contributes the YouTube-SBS dataset (7M+ stereo pairs from 423 videos). Results show improvements over image-based stereo conversion methods on L1 (11.76%), SSIM (6.39%), and PSNR (5.10%) vs. stereo-from-mono, at 5.6 FPS inference speed.

---

### Methodologies

* Dual-Branch Architecture: A disparity branch (pretrained DepthAnything) extracts depth-oriented features at 1/2 and 1/4 resolutions, while a context branch (stack of convolutions without aggressive downsampling) captures semantic and texture information.

* Implicit Disparity Guidance: Instead of relying on explicit, error-prone disparity maps, the model predicts a probability distribution of potential pixel shifts, serving as a differentiable relaxation of the warping operation.

* Layered Disparity Representation: 7 disparity layers with a mask selection algorithm allow pixel reuse across layers, avoiding blending artifacts and black holes from occlusion. Each layer contributes a warped image that is composited via learned masks.

---
### Youtube-SBS dataset

* Scale: It contains over 7 million stereo pairs from 423 high-resolution videos.

* Curation: Movie trailers, music videos, and game films; explicitly excludes 360° VR and gameplay with UI overlays.

* Validation: RAFT-based left-right consistency check. 71.27% of frames have <10% occluded area; most frames exhibit subtle stereo effects. Uses ε=4 threshold for flow consistency.

---
### Results

* Quantitative Improvement: ImmersePro outperforms previous state-of-the-art methods like "stereo-from-mono," improving results by 11.76% in L1, 6.39% in SSIM, and 5.10% in PSNR. All baselines are image-based methods.

* Efficiency: The model achieves a significantly higher inference speed of 5.609 FPS, compared to competitors that often run below 1.2 FPS.

**Compliance With Llm Reviewing Policy:**

Affirmed.

**Final Justification:**

My original review raised two primary concerns: (1) the absence of 3D geometric evaluation, and (2) the lack of perceptual/user evaluation for a task where viewing comfort is the ultimate criterion. The authors addressed both during the rebuttal. The MEt3R evaluation demonstrates 23.3% improvement in 3D structural consistency over the nearest baseline, directly following my suggestion to use DUSt3R-family methods. The pilot user study (n=7, Meta Quest 3) shows consistent gains across frame quality, temporal coherence, stereoscopic effects, and overall conformity, with a commitment to scale to 30+ participants in the revision. These additions meaningfully strengthen the paper's empirical foundation.

I have raised my score from 3 to 4 in light of these results. However, I remain unconvinced by the fundamental approach of learning stereo disparity without explicit camera geometry. The method produces perceptually plausible stereo effects, but without geometric grounding its utility is largely limited to VR content consumption.

**Key Questions For Authors:**

* Given that GT comparison is acknowledged as unreliable, could you provide a user study or perceptual evaluation?

* The fundamental limitation is geometric ambiguity: without camera parameters, output stereo pairs are uncalibrated and unusable for downstream 3D tasks. Any suggestions that we could include camera information?

**Limitations:**

* Please see the weakness and questions part.

**Strengths And Weaknesses:**

### Strengths

* End-to-End Video Focus: Unlike prior image-based methods, ImmersePro leverages spatial-temporal attention across frames. This is a meaningful architectural choice for stereo conversion where temporal consistency is critical for viewing comfort.

* Implicit disparity as differentiable relaxation: Predicting a probability distribution over disparity values rather than relying on explicit depth maps is well-motivated. The ablation convincingly shows the model fails to converge without this module, demonstrating its necessity.

* Performance: The model achieves an inference speed of 5.609 FPS, and improves L1 results and PSNR.

---

### Weaknesses

* Lack of 3D evaluation :  The method produces stereo video without known camera intrinsics or baseline distance, making the output geometrically uncalibrated. The authors acknowledge that "precise reproduction of the right view is impossible" without stereo parameters, yet no effort is made to evaluate geometric consistency. Modern 3D reconstruction methods (DUSt3R, MASt3R, VGGT) could verify whether predicted stereo pairs yield metrically consistent 3D structure. Without this, the output is purely a perceptual effect with no guarantee of downstream 3D utility.

* Unreliable GT and no perceptual evaluation: The authors acknowledge GT comparisons can be misleading due to focal length variance and averaging effects, yet L1/SSIM/PSNR remain the only quantitative evaluation. No user study or perceptual metric is provided for a task where human viewing comfort is the ultimate criterion.

---

> ### Author Rebuttal · Authors · 2026-03-30
>
> 1. **Lack of 3D evaluation**:  We thank the reviewer for suggesting using reconstruction-based methods to assess 3D structural consistency. Following this suggestion, we adopt MEt3R, a metric built upon DUSt3R, to evaluate the geometric consistency of the generated stereo pairs. As shown in the table below, our method achieves the best 3D structural consistency among all compared methods, outperforming the nearest competitor (Stereo From Mono) by **23.3\%**. We will add these results to our revised version.
>
>     |                 | 3D Photo | Stereo Diffusion | Stereo From Mono | Ours     |
>     |:----------------|:---------|:-----------------|:-----------------|:---------|
>     |         MEt3R ↓ | 11.97    | 9.52             | 6.90             | **5.29** |
>
> 2. **No perceptual evaluation**: We agree that perceptual evaluation is important. Due to time constraints during the rebuttal period, we present a pilot user study with only 7 participants (computer vision researchers), using a Meta Quest 3 headset. The study assessed four key dimensions on a 5-point Likert scale (higher is better):
>
>     - Frame Quality: Evaluated using generated right-eye frames only. Watch as 2D videos.
>     - Temporal Coherence: Evaluated via 3D stereo playback.
>     - Stereoscopic Effects: The strength of stereo effects.
>     - Overall Conformity: General visual experience from the user's perspective.
>
>     The study evaluates on four aspects, including frame quality, temporal coherence, stereoscopic effects, and overall conformity, in a 5-point Likert scale. We use a Meta Quest 3 headset for the evaluation. The users will first take a look at the generated right frames only for the frame quality measure, then play the videos in 3D left-right stereo mode for the other measures.
>
>     |                      | 3D Photo | Stereo Diffusion | Stereo From Mono | Ours     |
>     |:---------------------|:---------|:-----------------|:-----------------|:---------|
>     |  Frame Quality       | 2.57     | 3.57             | 3.28             | **4.28** |
>     | Temporal Coherence   | 2.28     | 3.00             | 3.57             | **4.14** |
>     | Stereoscopic Effects | 2.57     | 3.28             | 3.42             | **3.85** |
>     | Overall Comformity   | 2.14     | 3.28             | 3.57             | **4.14** |
>
>     Our method outperforms existing baselines across all categories. Notably, the gain in Temporal Coherence and Overall Conformity suggests that our approach provides a more stable and immersive viewing experience in a VR environment.
>
>     While this pilot study provides strong initial evidence of our method's superiority, we acknowledge the small participant pool. We will present a comprehensive user study with at least 30 participants in the revised version of the manuscript.
>
> 3. **Geometric ambiguity**: We agree that without known camera parameters, the generated stereo pairs are not metrically calibrated against the ground truth. However, we respectfully clarify that "uncalibrated" does not mean "unusable." Many reconstruction methods (such as DUSt3r) operate under unknown camera parameters and are usable for downstream 3D tasks. Thus, we argue that, though precise reproduction is difficult, visually plausible stereo conversion can be achieved without those parameters.
>
>     We acknowledge that for an exact reproduction of the right view (fully aligned with real-world ground truth), three core components are required:
>     1. Dense Depth Maps: To provide the underlying geometry.
>     2. Camera Parameters: To define the projection model.
>     3. Stereo-Specific Parameters: Such as the baseline and the Zero-Disparity Plane (ZDP), which are critical for reproducing the exact stereoscopic effects against the ground truth.
>
>     From an architectural perspective, the camera/stereo parameters could be integrated into the model using a cross-attention module, allowing the latent features to be conditioned on specific camera/stereo constraints.
>
>     However, the primary bottleneck remains data. Acquiring a large-scale, high-diversity dataset that includes synchronized stereo pairs alongside accurate camera and stereo metadata is exceptionally difficult. We believe that exploring these conditioning mechanisms on curated datasets is a promising avenue for future work.
>
>     We will include this discussion in our revised version.

---

> > ### Author Rebuttal · Reviewer_9xGx · 2026-04-02
> >
> > The authors have addressed both primary concerns with new quantitative evidence. The MEt3R evaluation directly follows my suggestion and demonstrates strong 3D structural consistency, outperforming the nearest baseline by 23.3%. The pilot user study, while limited to 7 participants, shows consistent improvements across all four perceptual dimensions on a VR headset.
> >
> > On geometric ambiguity, the clarification that uncalibrated stereo pairs remain usable for reconstruction is a fair point, and the suggested cross-attention conditioning mechanism is a reasonable future direction.
> >
> > I raise my score from 3 to 4. That said, I remain philosophically unconvinced by the overall approach: learning stereo disparity without explicit camera geometry feels like it sidesteps a fundamental problem rather than solving it. The method produces perceptually plausible stereo effects, but the lack of geometric grounding limits its utility beyond VR consumption.

---

> > > ### Author Response · Authors · 2026-04-05
> > >
> > > We thank the reviewer for raising the score. We agree that alternative problem formulations exist that are interesting to explore. Our approach is not based on explicit camera geometry. This is not a sidestep, but a deliberate focus on a different problem setting: stereo synthesis from unconstrained real-world data where camera and stereo parameters are absent. Our method is designed specifically for this scenario.

---

### Decision · Program_Chairs · 2026-04-30

**Decision:**

Accept (regular)

**Comment:**

This paper studies monocular-to-stereo video synthesis and proposes an end-to-end framework with implicit disparity learning. The reviewers found the problem interesting and the method technically reasonable. They also viewed the dataset contribution as useful, and found the improvements over prior methods meaningful.

The main concerns were about the lack of 3D evaluation, the absence of perceptual evaluation for a viewing-oriented task, the limited number of baseline comparisons, and the geometric ambiguity of learning stereo pairs without explicit camera parameters. In the rebuttal, the authors addressed these concerns with additional 3D consistency results, a pilot user study, more baseline comparisons, and clearer discussion of limitations. These additions improved the empirical support for the paper.

At the same time, some reservations remain about the underlying problem formulation, especially the lack of explicit geometric grounding and the resulting limits on broader 3D usefulness. Still, AC believes the paper has clear merit. On balance, the strengths outweigh the weaknesses, and AC therefore recommends weak acceptance.